# The rs508487, rs236911, and rs236918 Genetic Variants of the Proprotein Convertase Subtilisin–Kexin Type 7 (*PCSK7*) Gene Are Associated with Acute Coronary Syndrome and with Plasma Concentrations of HDL-Cholesterol and Triglycerides

**DOI:** 10.3390/cells10061444

**Published:** 2021-06-09

**Authors:** Gilberto Vargas-Alarcón, Oscar Pérez-Méndez, Héctor González-Pacheco, Julián Ramírez-Bello, Rosalinda Posadas-Sánchez, Galileo Escobedo, José Manuel Fragoso

**Affiliations:** 1Department of Molecular Biology, Instituto Nacional de Cardiología Ignacio Chávez, Mexico City 14080, Mexico; gvargas63@yahoo.com (G.V.-A.); opmendez@yahoo.com (O.P.-M.); 2School of Engineering and Sciences Campus CDMX, Tecnológico de Monterrey, Mexico City 14380, Mexico; 3Coronary Care Unit, Instituto Nacional de Cardiología Ignacio Chávez, Mexico City 14080, Mexico; hectorglezp@hotmail.com; 4Research Unit, Hospital Juárez de México, Mexico City 01460, Mexico; dr.julian.ramirez.hjm@gmail.com; 5Department of Endocrinology, Instituto Nacional de Cardiología Ignacio Chávez, Mexico City 14080, Mexico; rossy_posadas_s@yahoo.it; 6Unit of the Experimental Medicine, Hospital General de México, Dr. Eduardo Liceaga, Mexico City 06726, Mexico; gescobedog@msn.com

**Keywords:** genetics, susceptibility, acute coronary syndrome, proprotein convertase subtilisin–kexin type 7

## Abstract

Dyslipidemia has a substantial role in the development of acute coronary syndrome (ACS). Previous reports, including genome-wide associations studies (GWAS), have shown that some genetic variants of the proprotein convertase subtilisin–kexin type 7 *(PCSK7)* gene are associated with plasma lipid levels. In the present study, we evaluated whether *PCSK7* gene polymorphisms are significantly associated with the plasma lipid profile and ACS. Three *PCSK7* gene polymorphisms (rs508487 *T*/*C*, rs236911 *C*/*A,* and rs236918 *C*/*G*) were determined using TaqMan genotyping assays in a group of 603 ACS patients and 622 healthy controls. The plasma lipid profile was determined in the study groups by enzymatic/colorimetric assays. Under the recessive model, the rs236918 *C* allele was associated with a high risk of ACS (OR = 2.11, *pC* = 0.039). In the same way, under the recessive and additive models, the rs236911 *C* allele was associated with a high risk of ACS (OR = 1.95, *pC* = 0.037, and OR = 1.28, *pC* = 0.037, respectively). In addition, under the co-dominant model, the rs508487 *T* allele was associated with a higher risk of ACS (OR = 1.78, *pC* = 0.010). The *CCC* and *TCC* haplotypes were associated with a high risk of ACS (OR = 1.21, *pC* = 0.047, and OR = 1.80, *pC* = 0.001, respectively). The rs236911 *CC* and rs236918 *CC* genotypes were associated with lower high-density lipoproteins-cholesterol (HDL-C) plasma concentrations, whereas the rs236911 *CC* genotype was associated with a higher concentration of triglycerides, as demonstrated in the control individuals who were not receiving antidyslipidemic drugs. Our data suggest that the *PCSK7* rs508487 *T*/*C*, rs236911 *C*/*A,* and rs236918 *C*/*G* polymorphisms are associated with the risk of developing ACS, and with plasma concentrations of HDL-C and triglycerides.

## 1. Introduction

ACS is a complex multifactorial disorder resulting from genetics and various environmental factors, coupled with dyslipidemias, hypertension, diabetes, obesity, and a smoking habit—factors that play a fundamental role in the development and progression of atherosclerosis [1,2,3]. Dyslipidemias are considered one of the most important risk factors for acute coronary syndrome (ACS) and remain the primary target of current cardiovascular risk reduction strategies [2,3,4].

The proprotein convertase subtilisin–kexin type 7 (PCSK7) is a member of the proprotein convertases family. These proprotein convertases are related to the cleavage peptide bonds of various proteins and peptides in multiple subcellular compartments included in the secretory pathway [endoplasmic reticulum (ER), various Golgi cisternae, endosomes], which leads to the activation/inactivation of receptors, ligands, enzymes, viral glycoproteins, or growth factors [5]. Recent studies have shown that the human PCSK7 proprotein is implicated in the regulation of apolipoprotein A-5 (apoA-5) and is associated with a higher triglyceride concentration [6,7], both with an important role in the development of cardiovascular diseases [3,8,9,10]. The PCSK7 proprotein is encoded by the *PCSK7* gene, located in the q23.3 region of chromosome 11. Genome-wide association studies (GWAS) have shown that some single nucleotide polymorphisms (SNPs) of the *PCSK7* gene are associated with hypertriglyceridemia, with large effects on HDL-C and/or triglyceride levels [10,11,12,13]. Recent studies have associated three novel SNPs [*PCSK7* nearGene-3 rs508487 *T*/*C*, *PCSK7* intron 6 rs236911 *C*/*A,* and *PCSK7* intron 9 rs236918 *C*/*G*] with an over-expression of the PCSK7 and hypertriglyceridemia [6,10,11,12,13], as well as with the risk of developing atherosclerosis and liver disease [4,10].

In this context, considering the important role of the PCSK7 protein in the regulation of the Apo-A5 as well as its association with triglyceride levels, the present study aimed to establish the association of the *PCSK7* rs508487 *T*/*C*, rs236911 *C*/*A,* and rs236918 *C*/*G* polymorphisms with the susceptibility to developing ACS. Moreover, we evaluated whether these polymorphisms were associated with lipid profile plasma concentrations in a Mexican population sample.

## 2. Materials and Methods

### 2.1. Characteristics of the Study Population

The sample size for the unmatched case-control study was calculated with a power of 80% and an alpha error of 0.05. The total sample size required to carry out this study was 528 Mexican mestizo individuals (264 patients with ACS and 264 control individuals) (http://www.openepi.com/SampleSize/SSCC.html (accessed on 5 February 2020)). This study included 1225 Mexican mestizos—603 patients with ACS and 622 healthy controls. From July 2010 to 2015, 603 patients with ACS (82% men and 18% women, with a mean age of 58 ± 10.5 years) were referred to the Instituto Nacional de Cardiologia Ignacio Chavez. The diagnosis of ACS was made based on clinical characteristics, electrocardiographic changes, and biochemical markers of cardiac necrosis (creatinine kinase isoenzymes, creatinine phosphokinase, or troponin I above the upper limit of normal); the European Society of Cardiology (ESC) and American College of Cardiology (ACC) definitions were followed [14,15]. The control group included 622 healthy individuals (69% men and 31% women, with a mean age of 54.0 ± 7.7 years) that were recruited from the database cohort of the Genetics of Atherosclerosis Disease (GEA) Mexican study. The GEA study investigated the genetic factors associated with premature coronary artery disease (CAD), atherosclerosis, and other coronary risk factors in the Mexican population [16]. All subjects were asymptomatic and apparently healthy, without a family history of premature CAD or atherosclerosis; they were recruited from June 2009 to June 2013 from blood bank donors and with the assistance of brochures posted in social service centers. The exclusion criteria included congestive heart failure and liver, renal, thyroid, or oncological disease. Likewise, the control subjects had a coronary calcium score of “0” determined by computed tomography, indicating the absence of subclinical atherosclerosis in these individuals [16]. The contribution of the *PCSK7* rs508487 *T*/*C*, rs236911 *C*/*A,* and rs236918 *C*/*G* SNP genotypes on the lipid plasma levels was evaluated only in a subset of healthy controls. All the included subjects were ethnically matched and considered Mexican mestizos only if they and their ancestors (at last three generations) had been born in the country. The study complies with the Declaration of Helsinki and was approved by the Ethics and Research commission of Instituto Nacional de Cardiologia Ignacio Chavez. Written informed consent was obtained from all individuals enrolled in the study.

### 2.2. Laboratory Analyses

Cholesterol and triglyceride plasma concentrations were determined by enzymatic/colorimetric assays (Randox Laboratories Crumlin, County Antrim, UK). HDL-cholesterol (C) concentrations were determined after precipitation of the apo B-containing lipoproteins by the phosphotungstic acid-Mg^2+^ method. The low-density lipoprotein-cholesterol (LDL-C) concentration was determined in samples who had a triglyceride level lower than 400 mg/dL with the Friedewald formula [17]. Dyslipidemia was defined as one or more of the following characteristics: cholesterol > 200 mg/dL, LDL-C > 130 mg/dL, HDL-C < 40 mg/dL, or triglycerides > 150 mg/dL, according to the guidelines of the National Cholesterol Education Project (NCEP) Adult Treatment Panel (ATP-III)-(https://www.nhlbi.nih.gov/health-topics/all-publications-and-resources/third-report-expert-panel-detection-evaluation-and-0 (accessed on 7 May 2020)). Type 2 diabetes mellitus (T2DM) was defined by a fasting glucose ≥ 126 mg/dL and was also considered when participants reported glucose-lowering treatment or diagnosis of T2DM by a physician. Hypertension was defined by systolic blood pressure ≥ 140 mmHg and/or diastolic blood pressure ≥ 90 mmHg, or the use of oral antihypertensive therapy [16].

### 2.3. Genetic Analysis

DNA extraction was performed from peripheral blood in agreement with the method described by Lahiri and Nurnberger [18]. The *PCSK7* nearGene-3 rs508487 *T*/*C*, *PCSK7* intron 6 rs236911 *C*/*A*, and *PCSK7* intron 9 rs236918 *C*/*G* SNPs were genotyped using 5′ exonuclease TaqMan genotyping assays in a 7900HT Fast Real-Time PCR System, according to manufacturer’s instructions (Applied Biosystems, Foster City, CA, USA). To avoid genotyping errors, 10% of the samples were determined twice; the results were concordant for all cases.

### 2.4. Inheritance Model Analysis

The association of the rs508487 *T*/*C*, rs236911 *C*/*A,* and rs236918 *C*/*G* SNPs with ACS patients was performed under the following inheritance model: additive (major allele homozygotes versus heterozygotes versus minor allele homozygotes), codominant (major allele homozygotes versus minor allele homozygotes), dominant (major allele homozygotes versus heterozygotes + minor allele homozygotes), over-dominant (heterozygotes versus major allele homozygotes + minor allele homozygotes), and recessive (major allele homozygotes + heterozygotes versus minor allele homozygotes) using logistic regression, adjusting for cardiovascular risk factors. 

### 2.5. Analysis of the Haplotypes

The linkage disequilibrium analysis (LD, D″) and haplotype construction were performed using Haploview version 4.1 (Broad Institute of Massachusetts Institute of Technology and Harvard University, Cambridge, MA, USA).

### 2.6. Statistical Analysis

Gene frequencies of *PCSK7* polymorphisms on patients and controls were obtained by direct counting. The Hardy–Weinberg equilibrium was evaluated by the Chi^2^ test. The analysis of data was performed with SPSS version 18.0 (SPSS, Chicago, Il). The Mann–Whitney U test was used for the comparison of continuous variables between control and ACS groups. For categorical variables, Chi^2^ or Fisher’s exact tests were performed. The logistic regression analysis was used to test the association of the polymorphisms with ACS under inheritance models, adjusting according to cardiovascular risk factors. All *p*-values were corrected (*pC*) by the Bonferroni test. The values of *pC* < 0.05 were considered statistically significant, and all odds ratios (OR) were presented with 95% confidence intervals. The occurrence of the ACS in our study was based on the OR values: (a) OR = 1 does not affect the odds of developing ACS, (b) OR > 1 is associated with higher odds of developing ACS, and (c) OR < 1 is associated with lower odds of developing ACS. For the control subsets grouped by genotypes, the plasma lipid levels were expressed as means ± SD; comparisons were performed by ANOVA and least significant difference (LSD) as post hoc test. The *p* values < 0.05 were considered statistically significant. The statistical power used to detect an association with ACS was 0.80. We used the OpenEpi software [http://www.openepi.com/SampleSize/SSCC.html (accessed on 5 January 2021)].

## 3. Results

### 3.1. Characteristics of the ACS Patients and Controls

The anthropometric and biochemical parameters of the ACS patients and healthy controls are presented in Table 1. There were significant differences between the ACS patients and healthy controls. Compared to healthy controls, the ACS patients had a higher frequency of T2DM, hypertension, dyslipidemia, and a smoking habit. Conversely, the total cholesterol, triglycerides, and LDL-C levels in ACS patients were lower than those in the control group, probably due to statin intake and dietetic restrictions.

### 3.2. Association of PCSK7 SNPs with Plasma Lipids Levels

Previous reports have suggested the role of the PCSK7 on plasma lipid levels [6,8,10,11,12,13]. Therefore, in this study we first explored the potential association of rs508487 *T*/*C*, rs236911 *C*/*A*, and rs236918 *C*/*G* SNPs with total cholesterol, LDL-C, HDL-C, ratio LDL/HDL, and triglyceride plasma concentrations. We also looked for a possible association of these *PCSK7* SNPs with other cardiovascular risk factors (BMI, blood pressure, glucose). For this analysis, we selected only the group of healthy controls who were not receiving any drug that affected the plasma lipid profile (*n* = 622). We did not include the analysis of plasma lipid levels in patients with ACS because in the setting of coronary syndrome, these levels may be altered by using anti-dyslipidemic or anti-hypertensive drugs [19,20,21]. The analysis showed that the rs508487 *T*/*C,* rs236911 *C*/*A*, and rs236918 *C*/*G* SNPs were not associated with the following parameters: total cholesterol, LDL-C, BMI, blood pressure, ratio LDL/HDL, and glucose (Appendix A). However, we observed significant differences in HDL-C and triglyceride plasma levels when subjects were grouped according to these SNPs. In this context, subjects carrying the rs236918 *CC* genotype had a lower HDL-C plasma concentration (37.0 ± 8.2 mg/dL) than carriers of both *CG* and *GG* genotypes (43.8 ± 13.1 mg/dL, *p* = 0.041, and 45.0 ± 13.1 mg/dL, *p* = 0.015, respectively) (Figure 1A). On the other hand, individuals with the rs236911 *CC* genotype showed a lower concentration of HDL-C in plasma (37.6 ± 9.3 mg/dL) than individuals with either *CA* (43.8 ± 13.1 mg/dL, *p* = 0.025) or *AA* genotypes (45.1 ± 12.9 mg/dL, *p* = 0.004) (Figure 1B). In addition, individuals with the rs236911 *CC* genotype showed a higher concentration of triglycerides in plasma (224.4 ± 127.4 mg/dL) than individuals with either *CA* (177.7 ± 91.85 mg/dL, *p* = 0.032) or *AA* genotypes (171.8 ± 90.2 mg/dL, *p* = 0.023) (Figure 1C). The rs508487 T/C polymorphism was not associated with plasma lipid concentrations (data not shown).

### 3.3. Allele and Genotype Frequencies

The PCSK7 SNP associations with plasma lipid levels described above suggested that these SNPs could be associated with coronary heart disease. In this context, we explored this possibility in a group of patients with ACS and their corresponding controls. Allele and genotype frequencies of *PCSK7* polymorphisms in ACS patients and healthy controls are shown in Table 2. Observed and expected frequencies of the three polymorphisms were in Hardy–Weinberg equilibrium. The distribution of alleles and genotypes in patients with ACS and healthy controls showed significant differences in the 3 studied SNPs. For the genotypes (*p* = 0.008 for rs236918 *C*/*G*, *p* = 0.014 for rs236911 *C*/*A*, and *p* = 0.012 for rs508487 *T*/*C*), as well as for alleles (*p* = 0.010 for rs236918 *C*/*G*, *p* = 0.015 for rs236911 *C*/*A*, and *p* = 0.015 for rs508487 *T*/*C*).

### 3.4. Association of PCSK7 SNPs with ACS

Under the recessive model, the *CC* genotype of the rs236918 C/*G* polymorphism increased its risk of developing ACS (OR = 2.11, 95% CI: 1.02–4.37, *pC* = 0.039). In addition, under the recessive and additive models, the *CC* genotype of the rs236911 *C*/*A* was associated with a high risk of developing ACS (OR = 1.95, 95% CI: 1.03–3.67, *p*C_Recessive_ = 0.037, and OR = 1.28, 95% CI: 1.01–1.61, *p*C_Additive_ = 0.037). On the other hand, under the co-dominant model, the *CT* genotype of the rs508487 T/C polymorphism was associated with a high risk of developing ACS when compared to the *CC* genotype (OR = 1.78, 95% CI: 1.15–2.76, *pC* = 0.010) (Table 3). All models were adjusted for sex, age, blood pressure, body mass index (BMI), glucose, total cholesterol, HDL-C, LDL-C, triglycerides, ratio LDL-C/HDL-C, and smoking habit.

### 3.5. Linkage Disequilibrium Analysis

The linkage disequilibrium analysis showed a strong linkage disequilibrium (D’ > 0.85) between the rs508487 *T*/*C*, rs236911 *C*/*A*, and rs236918 C/*G* polymorphisms. In addition, this analysis showed three (*CAG, CCC,* and *TCC)* of four haplotypes with important differences between the two groups (Table 4). The *CAG* haplotype was associated with a low risk of developing ACS (OR = 0.83, 95% CI: 0.69–0.99, *pC* = 0.021), whereas the *CCC* and *TCC* haplotypes were associated with a high risk of developing ACS (OR = 1.21, 95% CI: 0.97–1.51, *pC* = 0.045 and OR = 1.80, 95% CI: 1.23–2.64, *pC* = 0.001, respectively).

## 4. Discussion

Dyslipidemias play an important role in the development of ACS, which is a multifactorial and polygenic disorder consequence of atherosclerosis. In the present work, we studied three polymorphisms (rs508487 *T*/*C*, rs236911 *C*/*A*, and rs236918 *C*/*G*) located on the *PCSK7* gene that encodes to the human PCSK7 proprotein. This protein has been implicated in the regulation of apoA-5 and has been associated with a higher triglyceride concentration [3,6,7,8,9]. In our study, we reported the association of the rs508487 *T*, rs236911 *C*, and rs236918 *C* alleles with the risk of developing ACS. To the best of our knowledge, this study is the first to describe the association between these polymorphisms and the presence of ACS. The association of these SNPs with cardiovascular diseases and other diseases in different populations is scarce and controversial. For example, in agreement with our data, Hoogeveen et al. reported that the *T* allele of the rs508487 polymorphism increased the risk of developing coronary heart disease in the CARDIoGRAM study; however, this association was not observed in the ARIC study [4]. In the same way, Kurano et al. reported in a GWAS that the rs508487 *T* and rs236911 *C* alleles were associated with the risk of hypertriglyceridemia in the Japan Pharmacogenomics Data Science Consortium (JPDSC) [13]. On the other hand, several studies have reported the association of the rs236918 *C*/*G* polymorphism with inflammatory diseases such as non-alcoholic fatty liver disease (NAFLD) [10] and liver cirrhosis [22,23], but not with cardiovascular diseases. In this context, Dongiovanni et al. reported that the rs236918 *C* allele was associated with the risk of developing NAFLD and higher triglycerides levels in the Caucasian population [10]. These results agree with our data. In the same way, Stickel et al. and Pelucchi et al. reported that the rs236918 *C* allele is associated with an increased risk of developing liver cirrhosis in the Caucasian population [22,23]. Additionally, in our study, we found that the *CCC* and *TCC* haplotypes were associated with a high risk of ACS, whereas the *CAG* haplotype was associated with a low risk. Interestingly, the risk haplotypes have the rs236911 *C* and rs236918 *C* alleles, which were both associated independently with the disease. This finding corroborated the role of these alleles in the genetic susceptibility of ACS, whether they were analyzed independently or as haplotypes. As can be seen in our study, the rs508487 *T*/*C*, rs236911 *C*/*A*, and rs236918 *C*/*G* polymorphisms were associated with the presence of ACS; however, information about these polymorphisms is scarce in other populations. We believe that the association of the *PCSK7* polymorphisms with ACS may also be due to the allelic distribution of these polymorphisms, which varies according to the ethnic origin of the study populations. In this context, data obtained from the National Center for Biotechnology Information revealed that the individuals from Los Angeles with Mexican Ancestry, Mexican mestizos, Caucasians, and Africans had a lower frequency of the rs508487 *T* allele (5, 4, 5, and 1%, respectively) as compared to Asians (16%). On the other hand, the Mexican mestizos, Caucasians, Africans, as well as individuals from Los Angeles with Mexican ancestry had a lower frequency of the rs236911 *C* allele (22, 14, 12, and 23%, respectively), whereas the Asian population had a high frequency of this allele (51%). Concerning the rs236918 genetic variant, the rs236918 *C* allele had a lower frequency in the individuals from Los Angeles with Mexican Ancestry, Mexican mestizos, Caucasians, and Africans (22, 19, 13, and 16%, respectively) as compared to Asians (49%) (https://www.ncbi.nlm.nih.gov/variation/tools/1000genomes/ (accessed on 25 November 2020)), (https://www.ensembl.org/index.htm (accessed on 25 November 2020)).

The associations of the *PCSK7* gene polymorphisms (rs508487 *T*/*C*, rs236911 *C*/*A*, and rs236918 *C*/*G*) with lipid plasma concentrations have been proposed as the mechanism that would explain the relationship between these polymorphisms and the higher risk of developing coronary heart disease, atherosclerosis, hypertriglyceridemia, and fatty liver [4,6,10,11,12,13]. In this context, DonGiovanni et al. reported that the rs236918 *C* allele was associated with an increase of PCSK7 protein, triglycerides, aminotransferases, and decreased HDL-C levels [10]. Likewise, they showed that PCSK7 regulates lipogenesis, fat accumulation, inflammation, and fibrogenesis in HepG2 cells [10]. In the same way, Hoogeveen et al. reported that the rs508487 *T* allele is associated with increased levels of small LDL-C, total cholesterol, and triglycerides, and an increased risk of atherosclerosis [4]. On the other hand, Kurano et al. reported that the rs508487 *T* and rs236911 *C* alleles were associated with higher triglyceride plasma levels [13]. Considering such evidence, we explored whether the PCSK7 polymorphisms are related to the plasma lipid levels. For this analysis, we did not include the patients with ACS since the lipid profile may be altered by the use of anti-dyslipidemic or some anti-hypertensive drugs in the setting of coronary syndrome [19,20,21]. Instead, we only considered the subset of control individuals who were not receiving any drug that could alter the plasma lipid levels. By these means, we avoided the confounding effect of such drugs in the statistical analysis. Our results showed that the rs236911 *CC* and rs236918 *CC* genotypes were associated with low HDL-C levels. Additionally, the rs236911 *CC* genotype was associated with higher triglyceride levels. As far as we know, the precise mechanism by which PCSK7 decreases HDL-C and/or increases triglyceride levels in adverse events such as ACS, hypertriglyceridemia, and liver disease remains to be elucidated. Nonetheless, we suggest that the association of *PCSK7* polymorphisms with plasma lipid levels is related to the regulation of the apoA-5 levels by PCSK7 [6]. In this context, recent experimental studies have shown that the over-expression of PCSK7 increased the intracellular degradation of apoA-5 in the endoplasmic reticulum [6]; since apoA-5 is implicated in triglyceride metabolism, an increased degradation of the protein induced by PCSK7 would result in higher triglyceride plasma levels [10,11,12,13]. Furthermore, increased triglyceride plasma levels are a common feature determining HDL-C plasma levels. Similarly, transgenic mice deficient in the human apoA-5 had strongly elevated triglyceride levels [24]. Alternatively, the PCSK7 could regulate lipid concentrations due to its participation in iron metabolism via the shedding of the human transferrin receptor-1 (hTfR1) [25,26,27]. This receptor is associated with inflammation and oxidative stress, processes that lead to atherosclerosis and other cardiovascular diseases [28]. Considering our data, future investigations are warranted to understand the contribution of the *PCSK7* polymorphism to lipid concentrations.

We recognize that our study has some limitations that merit further consideration; the number of carriers of some polymorphisms and haplotypes is limited for statistical analysis. Notably, the number of included men was almost five times that of women in the group of ACS patients, and they were not matched by gender and age with the control group. Considering these limitations, the effect of the SNPs in plasma lipid levels should be taken carefully, and studies involving a larger number of individuals would be necessary to corroborate this association.

In summary, this study demonstrated that the rs508487 *T*/*C*, rs236911 *C*/*A*, and rs236918 *C*/*G* polymorphisms of the *PSCK7* gene are associated with the risk of developing ACS in a Mexican population. Moreover, it was possible to distinguish two haplotypes (*CCC* and *TCC*) associated with an increased risk of developing ACS. There was a statistically significant association of the rs236911 *C*/*A* and rs236918 *C*/*G* polymorphisms with lower HDL-C plasma levels, as well as an increase in triglyceride plasma levels associated with the rs236911 *C*/*A* SNP. Even if these results suggest a link between *PCSK7 *polymorphisms and the incidence of ACS via plasma lipid levels, the fact that mechanisms other than those related to lipoprotein metabolism may explain the association between these SNPs and ACS cannot be totally excluded.

Finally, based on our results and considering the specific genetic characteristics of the Mexican population, we propose that additional studies in a larger number of individuals have to be undertaken. These studies could help define the true role of these polymorphisms as markers of risk or protection in the development of ACS and other cardiovascular events.

## Figures and Tables

**Figure 1 cells-10-01444-f001:**
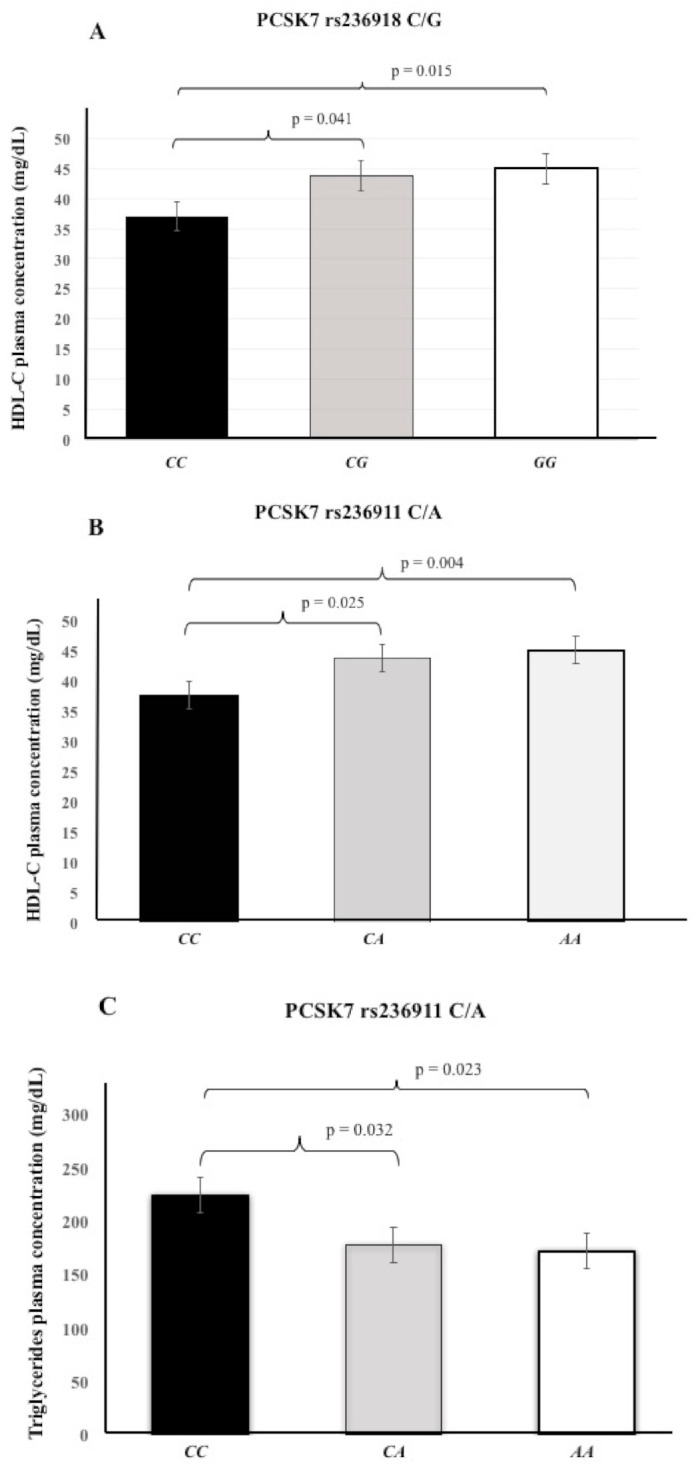
Genetic contribution of the *PCSK7* rs236911 *C*/*A, PCSK7* rs236918 *C*/*G,* and *PCSK7* rs508487 *T*/*C* polymorphisms on plasma lipid concentrations. (**A**) The rs236918 *CC* genotype showed low HDL-C levels when compared to the *CG*/*GG* genotypes (*p* = 0.041 and 0.015, respectively). (**B**) The rs236911 *CC* genotype showed low HDL-C levels when compared to the *CA* (*p* = 0.025) and *AA* (*p* = 0.004) genotypes. (**C**). In addition, the rs236911 *CC* genotype showed higher triglyceride levels when compared to the *CA* (*p* = 0.032) and *AA* (*p* = 0.023) genotypes.

**Table 1 cells-10-01444-t001:** Demographic, clinical, and biochemical parameters of the studied individuals.

Characteristics		ACS Patients (*n* = 603)	Healthy Controls (*n* = 622)	*p*-Value
		Median (percentile 25–75)	Median (percentile 25–75)	
Age (years)		58 (51–65)	54 (49–59)	0.001
Sex *n* (%)	Male	491 (82)	430 (69)	<0.001
	Female	112 (18)	192 (31)	
BMI (kg/m^2^)		27 (25–29)	28 (26–31)	0.593
Blood pressure (mmHg)	Systolic	130 (115–144)	115 (106–126)	<0.001
	Diastolic	80 (70–90)	72 (67–78)	<0.001
Glucose (mg/dL)		158 (102–188)	97 (84–99)	<0.001
Total cholesterol (mg/dL)		163 (127–200)	188 (164–210)	<0.001
HDL-C (mg/dL)		37 (32–45)	42 (35–53)	<0.001
LDL-C (mg/dL)		103 (75–133)	115 (94–134)	<0.001
Triglycerides (mg/dL)		149 (109–201)	154 (113–209)	0.144
Ratio LDL/HDL		2.72 (1.99–3.45)	2.73 (2.08–3.42)	0.891
		*n* (%)	*n* (%)	
Hypertension	Yes	345 (57)	182 (29)	<0.001
Type II diabetes mellitus	Yes	209 (35)	58 (9)	<0.001
Dyslipidemia	Yes	515 (85)	451 (72)	<0.001
Smoking	Yes	214 (35)	135 (22)	<0.001

Data are expressed as median and percentiles (25th–75th). *p* values were estimated using the Mann–Whitney U test continuous variables and the Chi-square test for categorical values. ACS: Acute coronary syndrome.

**Table 2 cells-10-01444-t002:** Allele and genotype distribution of *PCSK7* gene polymorphisms in ACS patients and healthy controls.

Polymorphic Site		ACS *n* = 603 (*n* [%])	Controls *n* = 622 (*n* [%])	** p*
*PCSK7* rs236918 *C*/*G*				
	Allele			
	*G*	926 (77)	1009 (81)	0.010
	*C*	280 (23)	235 (19)	
	Genotype			
	*GG*	354 (58.7)	403 (64.8)	
	*GC*	218 (36.1)	203 (32.6)	0.008
	*CC*	31 (5.1)	16 (2.6)	
*PCSK7* rs236911 *C*/*A*				
	Allele			
	*A*	886 (73)	967 (78)	0.015
	*C*	320 (27)	277 (22)	
	Genotype			
	*AA*	323 (53.6)	370 (59.5)	
	*AC*	240 (39.8)	227 (36.5)	0.014
	*CC*	40 (6.6)	25 (4)	
*PCSK7 UTR’3* rs508487 *T*/*C*				
	Allele			
	*C*	1127 (93)	1191 (96)	0.015
	*T*	79 (7)	53 (4)	
	Genotype			
	*CC*	524 (86.9)	569 (91.5)	
	*CT*	79 (13.1)	53 (8.5)	0.012
	*TT*	0	0	

Data are shown as *n* and frequency. * chi-square test.

**Table 3 cells-10-01444-t003:** Association of *PCSK7* SNPs with ACS.

		*n* (Genotype Frequency)		MAF	Model	OR (95%CI)	*pC*
*PCSK7*	rs236918 *C/G*						
Control	*GG*	*GC*	*CC*	*C*			
(*n* = 622)	403 (0.648)	203 (0.326)	16 (0.026)	0.188	Co-dominant	2.19 (1.05–4.58)	0.093
					Dominant	1.20 (0.91–1.58)	0.209
ACS	354 (0.587)	218 (0.361)	31 (0.051)	0.232	Recessive	2.11 (1.02–4.37)	0.039
(*n* = 603)					Over-dominant	1.07 (0.81–1.42)	0.639
					Additive	1.24 (0.98–1.58)	0.076
*PCSK7*	rs236911 *C/A*						
Control	*AA*	*AC*	*CC*	*C*			
(*n* = 622)	370 (0.595)	227 (0.365)	25 (0.040)	0.222	Co-dominant	2.08 (1.09–3.95)	0.065
					Dominant	1.25 (0.95–1.64)	0.121
ACS	323 (0.536)	240 (0.398)	40 (0.066)	0.265	Recessive	1.95 (1.03–3.67)	0.037
(*n* = 603)					Over-dominant	1.10 (0.83–1.46)	0.509
					Additive	1.28 (1.01–1.61)	0.037
*PCSK7 UTR’3*	rs508487 *T/C*						
Control	*CC*	*CT*	*TT*	*T*			
(*n* = 622)	569 (0.915)	53 (0.085)	0 (0.0)	0.042			
					Co-dominant	1.78 (1.15–2.76)	0.010
ACS	524 (0.869)	79 (0.131)	0 (0.0)	0.065			
(*n* = 603)							

ACS, acute coronary syndrome; MAF, minor allele frequency; OR, odds ratio; CI, confidence interval; *pC*, *p*-value. The *p*-values were calculated by logistic regression analysis, and ORs were adjusted for sex, age, blood pressure, BMI, glucose, total cholesterol, HDL-C, LDL-C, triglycerides, and smoking habit.

**Table 4 cells-10-01444-t004:** Frequencies of *PCSK7* haplotypes in patients with ACS and in healthy controls.

rs508487	rs236911	rs236918	ACS (*n* = 603)	Controls (*n* = 622)	OR	95%CI	*pC*
Haplotype			Hf	Hf			
*C*	*A*	*G*	0.730	0.765	0.83	0.69–0.99	0.021
*C*	*C*	*C*	0.167	0.142	1.21	1.00–1.51	0.045
*T*	*C*	*C*	0.062	0.036	1.80	1.23–2.64	0.001
*C*	*C*	*G*	0.035	0.040	0.88	0.57–1.33	0.312

Abbreviations: ACS: acute coronary syndrome; Hf = Haplotype frequency, *pC* = *p-corrected.* The order of the polymorphisms in the haplotypes is according to the positions in the chromosome (rs508487, rs236911, rs236918). Bold numbers indicate significant associations.

## Data Availability

The data presented in this study are available upon request from the corresponding author.

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
