# Peer review of "The rs508487, rs236911, and rs236918 Genetic Variants of the Proprotein Convertase Subtilisin–Kexin Type 7 (PCSK7) Gene Are Associated with Acute Coronary Syndrome and with Plasma Concentrations of HDL-Cholesterol and Triglycerides"

_cells, 2021, doi:10.3390/cells10061444_

Round 1

Reviewer 1 Report

This is to my knowledge the first study investigating genetic polymorphisms on PCSK7 re. CVD. The data are sound and the methods well described. I only have some minor issues that should be addressed:

1)The second sentence of the abstract should be corrected for English language "have been showed that > have shown that" and "variants genetics > genetic variants"

2) Since the association between PCSK9 and lipids have been shown previously, I suggest the authors start the results section with these results and then show the association with CVD

3) Could the authors tone sown their conclusion by stating that " it cannot be totally ruled out that other mechanisms that those related to lipoprotein metabolism may explain the association between these SNPs and CVD"

Reviewer 2 Report

In this work, the association between specific genetic variants of PCSK7 with plasma concentration of HDL and Tgs is investigated. The authors have shown that rs236911CC and rs23691CC genetoypes are associated with lower LDL while rs236911CC is associated with higher TG. Similar associations with other variants have been shown as well. The study was conducted with samples from more than 1000 human subjects.

1. The introduction section is rather short. It would be nicer to add more information to motivate the study and present the study in the context of wider literature. For example, multiple studies have shown that serum PCSK9 levels is predictive of occurrence (under certain conditions) and PCSK9 inhibitors have been used as preventative measure against ACS (lowering lipoprotein concentrations). It would be nice to have additional context.

2. The sex ratio in the study is very skewed – particularly with ACS patients. This should be addressed in the discussion as a limitation of study.

3. The authors mention that lower LDL levels in ACS patients (compared to control) may be due to statins intake (and dietary differences). Does statin not increase HDL levels in the plasma? How was this accounted for in the study? Was this addressed in statistical analysis? Since the genetic association with HDL levels are investigated here effect of medications would be confounding factor. What are other potential complications due to medications in ACS patients? This should be discussed in detail as well.

Round 2

Reviewer 2 Report

Authors have addressed my concerns.

One minor issue - Please fix the link to openepi website in proofs.

"http://www.openepi.com/SampleSize/SSCC.html" this is incorrect.

The correct link is  http://www.openepi.com/SampleSize/SSCC.htm (note that it ends with htm not html).